# Serum Glucose-6-Phosphate Dehydrogenase Activity as a Biomarker for Gastric Cancer Stage Prediction

**DOI:** 10.3390/cancers17233798

**Published:** 2025-11-27

**Authors:** Chang-Hwan Yeom, Jiewon Lee, Keun-Joo Bae, Kangseok Kim, Jongsoon Choi, Myeong-Hun Lim

**Affiliations:** 1 Department of Family Medicine, YCH Hospital, Seoul 05700, Republic of Korea; lymphych@hanmail.net (C.-H.Y.);; 2Department of Surgery, Bangre Hospital, Incheon 22228, Republic of Korea; 3Department of Family Medicine, Kosin University College of Medicine, Busan 49267, Republic of Korea

**Keywords:** G6PD, gastric cancer, biomarker, cancer stage

## Abstract

Gastric cancer is the fifth most prevalent cancer diagnosed worldwide. Since early identification of recurrence can enhance overall survival in gastric cancer, it is essential to develop a biomarker suitable for serial monitoring and precise prediction of cancer stage. This study aimed to evaluate the potential of glucose-6-phosphate dehydrogenase activity as a biomarker for predicting the stage of gastric cancer. Serum G6PD activity was markedly elevated in patients with advanced-stage disease compared with those in the early stage. In addition, ROC curve analysis demonstrated an AUC of 0.70, suggesting a fair diagnostic performance. This study highlighted the potential clinical value of serum G6PD activity as an indicator for assessing gastric cancer stage.

## 1. Introduction

Gastric cancer emerged as the fifth most frequently diagnosed malignancy worldwide, with an estimated 1.09 million new cases reported [1]. Notably, East Asia contributed the largest proportion, with approximately 660,000 cases, corresponding to an age-standardized incidence rate of 22.4 per 100,000 which is the highest among all regions [2]. Therefore, ensuring effective management and long-term surveillance of gastric cancer patients in East Asia has become a key clinical concern.

The 5-year recurrence rate of gastric cancer is reported to be 46.5%. Among recurrent cases, hematogenous metastasis accounts for 39.7%, while peritoneal dissemination occurs in 44.3% [3]. Moreover, gastric cancer shows a marked difference in 5-year survival rates depending on the stage at diagnosis. The 5-year survival rate exceeds 90% in stage I, but decreases to 85.4% and 70.2% in stages II and III. In stage IV gastric cancer, the survival rate drops to 29% [4]. Given the significantly poorer prognosis in advanced stages and high recurrence rate, early detection of recurrence during the asymptomatic phase has been reported to improve overall survival [5]. This approach may serve as a valuable strategy to enhance outcomes, particularly in patients with higher stage gastric cancer.

To enable early detection of recurrence and timely intervention after gastrectomy, endoscopic surveillance and abdominal computed tomography (CT) are commonly utilized [6]. According to a survey conducted by Korean Gastric Cancer Association, most institutions reported performing endoscopic follow-up at intervals of 6 to 12 months following gastrectomy [7]. In addition, the guidelines of the Japanese Gastric Cancer Association recommend performing endoscopy and either abdominal ultrasonography or abdominal computed tomography at 1- or 2-year intervals following gastrectomy [8]. Despite their widespread use, the cost of endoscopy and CT could be financially burdensome depending on the countries and the extent of insurance coverage [9]. Furthermore, endoscopy may be associated with adverse effect including sedation-related complication and perforation [10]. These limitations highlight the need for accessible, cost-effective and non-invasive biomarkers for disease monitoring.

The progression, recurrence, prognosis of gastric cancer are influenced by complex factors, including tumor microenvironment elements such as crosstalk between cancer-associated fibroblasts and myeloid cells, dysregulation of key signaling pathways like the PI3K/Akt/mTOR pathway, and specific genetic alterations such as the FAT4 gene mutation [11,12,13,14]. Additionally, factors such as histological subtype have also been reported to impact the prognosis and recurrence patterns of gastric cancer [15]. Because of complex pathogenic mechanisms and heterogeneous genetic mutations, developing a reliable single biomarker that can accurately predict the stage and monitor the therapeutic response of gastric cancer patients remains challenging. Therefore, it is essential to identify a simple, non-invasive, and reliable biomarker to provide efficient diagnosis strategies for gastric cancer patients.

Glucose-6-phosphate dehydrogenase (G6PD), which plays an essential role in maintaining intracellular NADPH homeostasis, has recently emerged as a promising biomarker due to its association with cancer stage and progression [16,17,18]. Notably, in Merkel cell carcinoma, a rare neuroendocrine skin malignancy, serum G6PD activity increases significantly with cancer stage and has been established as a reliable prognostic biomarker [19]. G6PD supports rapid cancer cell proliferation via the pentose phosphate pathway by producing NADPH, ribose-5-phosphate, erythrose-4-phosphate, essential for antioxidant defense, as well as nucleic acid and amino acid biosynthesis [20]. Therefore, this study aimed to investigate the potential of serum G6PD activity as a biomarker for predicting the stage of gastric cancer. We compared serum G6PD activity across difference cancer stages, performed ROC curve analysis and assessed its predictive performance.

## 2. Methods

### 2.1. Patients

A total of 64 patients were included in this cross-sectional study. All participants were adults aged 19 years or older who were diagnosed with gastric cancer and visited YCH Hospital between October 2018 and June 2025. Patients who did not undergo serum G6PD activity test, had unknown gastric cancer stage, were diagnosed with two or more types of cancer, or had any acute conditions including acute infections at the time of blood sampling were excluded. In addition, patients with known G6PD deficiency or those diagnosed with G6PD deficiency were also excluded. In this study, variables including gender, age, cancer stage, and G6PD activity were retrospectively collected from medical records. Furthermore, cancer stage was assessed at the time the G6PD activity test was performed.

### 2.2. G6PD Activity Test

G6PD activity is measured using the Dr. Rappeler G6PD system, a clinical diagnostic device supplied by Rappeler Company, located in Anyang, Republic of Korea, and approved by the Korean Ministry of Food and Drug Safety. Its quality control procedures and performance characteristics have been rigorously evaluated and validated by these regulatory authorities. According to the Instructions for Use (IFU), the device demonstrates regulator-approved performance and validated quality control parameters. Specifically, repeatability and reproducibility testing showed coefficients of variation ranging from 3.81% to 18.05% and 3.92% to 16.61%, respectively. Accuracy and correlation analyses revealed a strong agreement with a comparative product, with a correlation coefficient of 0.99. Analytical sensitivity was also verified, with a measurement range of 0–300 U/dL, a limit of detection of 2 U/dL, a limit of blank of 0 U/dL, and a limit of quantitation of 17 U/dL.

The device quantitatively measures total G6PD enzyme activity in blood, as well as G6PD activity normalized to hemoglobin levels. The measurement is performed using R1 analyzer, a CE-certified analytical device based on an electrochemical method. A drop of whole blood is applied to a G6PD test strip, where the current generated during the reduction of NADP+ to NADPH in the blood is measured. G6PD catalyzes this reaction, and the resulting NADPH reduces ferricyanide present in the reagent. The amount of current generated is proportional to the amount of NADPH produced, thereby allowing indirect quantification of G6PD activity. In this study, G6PD activity was expressed as the activity per unit of hemoglobin (g/dL), calculated by dividing the total G6PD activity by the hemoglobin concentration measured using the same device.

### 2.3. Statistical Analysis

First, variables were compared between early-stage and advanced-stage gastric cancer. In this study, early-stage gastric cancer was defined as stage I–II and advanced-stage gastric cancer was defined as stage III–IV based on TNM staging. The classification is commonly used in biomarker studies to distinguish between early and advanced disease to reflect differences in clinical characteristics [21,22]. A chi-square test was used to compare gender distribution between two groups. As the continuous variables showed skewed distributions, the Mann–Whitney U test was performed to compare age and G6PD activity according to cancer stage. Second, to evaluate whether the independent variables were predictive factors for gastric cancer stage, univariate and multivariable logistic regression analyses were conducted. The odds ratio (OR) of G6PD activity for gastric cancer stage was calculated, and its statistical significance was assessed. Third, a receiver operating characteristic (ROC) curve analysis was performed to assess the ability of G6PD activity to predict cancer stage, and the optimal cutoff point was identified. Finally, sensitivity, specificity, positive predictive value (PPV) and negative predictive value (NPV) were calculated based on the identified cutoff point. In addition, a sensitivity analysis was performed after stratifying the patients by gender. To compare G6PD activity within each gender subgroup, the Mann–Whitney U test was conducted. ROC curve analyses were also performed, and AUC values were calculated for both men and women. Statistical analyses were conducted using R software (version 4.4.2). All tests were two-tailed, and statistical significance was defined as a *p* value less than 0.05.

## 3. Results

### 3.1. General Characteristics

This cross-sectional study included a total of 64 patients. In this study, gastric cancer was first divided into early-stage (Stage I–II) and advanced-stage (Stage III–IV). Variables were compared between the early-stage and advanced-stage gastric cancer groups, and the results are presented in Table 1. In the early-stage gastric cancer group, 29.6% of patients were male, whereas in the advanced-stage group, 59.5% were male. A chi-square test revealed a statistically significant difference in the distribution of gender to cancer stage. No significant difference in age was observed between the two groups classified by cancer stage, as determined by Mann–Whitney U test. The median G6PD activity was significantly higher in the advanced-stage group (12.4 U/g Hb) compared to the early-stage group (10.4 U/g Hb). Figure 1 shows the distribution of G6PD activity according to cancer stage, demonstrating a significantly higher median value in the advanced-stage gastric cancer compared to the early-stage gastric cancer.

### 3.2. Logistic Regression Analysis for Predictors of Gastric Cancer Stage

Logistic regression analysis was conducted to evaluate whether G6PD activity was independently associated with cancer stage. Univariate logistic regression analysis was performed with age, gender, and G6PD activity as independent variables and cancer stage as dependent variable. Among the independent variables, only G6PD activity was significantly associated with cancer stage. In the multivariable logistic regression model adjusting for age and gender, G6PD activity remained an independent predictor of advanced-stage gastric cancer (OR 5.62, 95% CI 1.41–22.38).

### 3.3. Receiver Operating Characteristic Curve for G6PD Activity

As shown in Figure 2, diagnostic performance was assessed using a receiving operating characteristic (ROC) curve, and the area under the curve (AUC) was calculated. The AUC was 0.70 (95% CI, 0.57–0.83). Youden’s index was applied to identify the optimal cutoff point of G6PD activity that best discriminates between early-stage and advanced-stage gastric cancer. Youden’s index, defined as sensitivity + specificity − 1, indicates the cutoff point with the best balance between sensitivity and specificity [23]. The optimal cutoff point calculated in this study was 12.05 U/g Hb.

In Table 2, sensitivity, specificity, positive predictive value (PPV) and negative predictive value (NPV) were calculated for five different cutoff points. Considering both sensitivity and specificity, the most appropriate cutoff value was 12.05 U/g Hb. Using a cutoff of 12.05 U/g Hb, G6PD activity demonstrated a sensitivity of 0.59 and a specificity of 0.78 in predicting advanced-stage gastric cancer.

### 3.4. Sensitivity Analysis

After stratifying by gender, male patients showed significantly higher G6PD activity in the advanced-stage group compared with the early-stage group (Appendix A). In female patients, the median G6PD level was higher in the advanced-stage group than in the early-stage group, although the difference was not statistically significant. In the ROC analysis, the AUC was 0.830 in male and 0.593 in female, indicating better discriminatory performance in men (Appendix A).

## 4. Discussion

The findings of this study highlight the potential clinical utility of G6PD activity as a biomarker for distinguishing between early-stage and advanced-stage gastric cancer. G6PD activity was significantly higher in the advanced-stage group compared to early-stage group and served as a predictive factor for gastric cancer stage. Furthermore, ROC curve analysis revealed an AUC of 0.70, indicating that the serum G6PD activity has fair diagnostic performance and clinical utility [24]. The optimal cutoff point for predicting stage of gastric cancer was 12.05 U/g Hb, with a sensitivity of 0.59 and a specificity of 0.78. In sensitivity analysis, the AUC values were 0.830 for men and 0.593 for women. This is likely attributable to the small number of female patients (N = 34), which limits the ability to detect significant differences in G6PD activity or to adequately assess its diagnostic performance. Future studies with a sufficiently large number of both male and female participants will be necessary to validate the findings. Previous studies indicate that serum G6PD activity elevates as cancer progresses and suggest the potential utility of G6PD activity as a tumor marker reflecting cancer stage, which aligns with the findings of this study. Serum G6PD activity significantly increases with cancer stage in Merkel cell carcinoma and a notable decrease in G6PD activity has been observed following surgical resection or radiation therapy [19]. In addition, another previous study reported that G6PD overexpression in gastric cancer tissues was associated with the TNM stage and the presence of distant metastasis [25].

G6PD is gaining attention as a promising biomarker as its biological mechanisms involved in cancer cell proliferation are progressively being elucidated [16]. Firstly, G6PD acts as the rate-limiting enzyme in the Pentose Phosphate Pathway (PPP), one of two crucial pathways, along with aerobic glycolysis, for cancer cell proliferation [26,27,28]. Unlike normal cells, cancer cells tend to generate ATP through rapid glycolysis, even in the presence of sufficient oxygen, without utilizing the oxidative phosphorylation pathway following glycolysis [29]. This phenomenon is known as the Warburg effect, and cancer cells maintain a high glucose uptake rate to facilitate this rapid glycolysis [29]. Lactate and pyruvate, the end products of glycolysis, induce the accumulation of HIF-1α and promote the activation of G6PD [30,31]. Furthermore, G6PD activation has been reported to be essential for maintaining aerobic glycolysis [32]. Accordingly, cancer cells activate both glycolysis and the PPP through various mechanisms of metabolic remodeling.

Meanwhile, the PPP plays a protective role within rapidly proliferating cancer cells. The elevated metabolic activity of cancer cell leads to the accumulation of high levels of reactive oxygen species (ROS). The PPP is essential for the generation of NADPH, which reduces glutathione and thereby helps decrease intracellular oxidative stress [33]. Furthermore, the PPP also plays a crucial role in supplying precursors for RNA and amino acids, both of which are essential for cell division [33]. Thus, G6PD, central to the PPP, is an indispensable enzyme for cancer cell proliferation, working in conjunction with glycolysis.

In addition, G6PD promotes gastric cancer cell proliferation and metastasis through mechanisms mediated by NF-kB and hexokinase 2 (HK2). Increased G6PD expression activates NF-kB, a protein complex that regulates the expression of inflammation-related cytokines [34]. Moreover, G6PD promotes the production of hepatocyte growth factor (HGF) from tumor mesenchymal stem cells, which regulate tumor microenvironment surrounding cancer cells, via NF-kB activation [35]. HGF produced by tumor mesenchymal stem cells increases HK2 expression in nearby gastric cancer cells, and HK2 plays an essential role in gastric cancer cell proliferation, migration, and gastric cancer progression [35]. Therefore, G6PD participates in gastric cancer progression through multiple mechanisms, and it is indispensable for the proliferation and migration of gastric cancer cell.

Various liquid biopsy biomarkers have been revealed for predicting and evaluating the stage of gastric cancer [36,37]. First, serum exosomal miRNAs such as miR-19b-3p represent potential biomarkers for both the diagnosis and staging of gastric cancer. Exosomal miRNAs are a class of RNAs that regulate mRNA expression within cells, and the expression levels can be measured after isolating exosomes from the serum. Consistent with the findings of the present study, expression levels of exosomal miRNAs showed significantly higher levels in stage III–IV compared to stage I–II [38]. In addition, miRNAs were evaluated solely for their diagnostic utility in gastric cancer, and the reported AUC values for the two miRNAs were 0.786, 0.769, respectively. However, exosomal miRNA expression must be quantified through multiple steps, including serum exosome isolation, miRNA extraction, and real-time PCR, which limits its accessibility and requires substantial processing time. Furthermore, the detection of circulating tumor cells (CTCs) and disseminated tumor cells (DTCs) serves as one of the representative biomarkers for predicting cancer recurrence, metastasis, and staging. CTCs refer to tumor cells present in the bloodstream, whereas DTCs are tumor cells residing in the bone marrow. These cells can be detected using techniques such as RT-PCR or immunocytochemistry. The detection rate of tumor cells is significantly higher in stage III–IV gastric cancer compared to stage I–II, and was also elevated in patients with metastasis compared to those without metastasis [39]. Additionally, for the diagnosis of gastric cancer, circulating tumor cell (CTC) detection has been reported to show a sensitivity of 85.3% and a specificity of 90.3% [40]. Currently, the detection of CTCs is predominantly performed using RT-PCR to identify cancer-specific gene expression. However, a major limitation of this method is its suboptimal specificity, as the target genes may also be expressed in normal cells [39]. Additionally, for both methods, standardized protocols have not yet been established, and neither cutoff points for stage prediction nor diagnostic performance have been thoroughly investigated. Moreover, for the diagnosis of gastric cancer, miRNAs and CTCs showed higher AUC, sensitivity, and specificity compared to serum G6PD levels. However, biomarkers that have been evaluated for their diagnostic performance in predicting gastric cancer stage are scarce, highlighting the advantage of serum G6PD levels as a potential tool to screen for disease progression in gastric cancer patients. In addition, the serum G6PD activity test offers advantages in accessibility and reproducibility, as it employs a standardized measure protocol and yields rapid results from just a single drop of blood.

Considering that the model showed an AUC of 0.70 with moderate sensitivity and specificity, the result suggests a potential role of serum G6PD activity as a biomarker for gastric cancer stage, but the predictive performance is limited for immediate clinical application. Further validation in larger, independent cohorts will be necessary to confirm findings in this study and enhance generalizability. Therefore, while our study provides preliminary evidence supporting the association between serum G6PD levels and gastric cancer stage, additional studies are required to establish clinical utility of serum G6PD test.

There are multiple strengths in this study. Firstly, this study is the first to investigate and establish the diagnostic performance and cutoff point of serum G6PD activity as a potential biomarker for gastric cancer. Extending beyond the established biological mechanisms of G6PD activity in cancer progression, we quantitatively assessed serum G6PD activity and investigated its feasibility for clinical application. Secondly, this study employed a single method for measuring G6PD activity to enhance reproducibility. Thirdly, the G6PD activity assay used in this study is non-invasive and provides results within a short time of 4 min, allowing for repeated measurement.

Nonetheless, this study has certain limitations that should be acknowledged. At first, the present study did not account for various clinical variables, including underlying comorbidities. Although G6PD activity is known to be influenced by various factors such as infection and diabetes mellitus, only age and gender were included as covariates in the analysis due to limitations in the available clinical records. Future research should include adjustments for potential confounders, such as the presence of diabetes mellitus and ECOG performance status, to validate the predictive value of G6PD activity for gastric cancer staging. Secondly, this study was conducted at a single center using retrospective medical records, which may introduce selection bias and limit the generalizability of the findings. External validation cohorts which include a balanced representation of patient across diverse clinical backgrounds are necessary to confirm the reproducibility of the proposed biomarker across different populations. Third, although the biological association between G6PD expression in tumor tissues and cancer progression has been well established, the underlying mechanisms linking serum G6PD activity to tumor progression remain poorly understood. Therefore, further studies are needed to elucidate the mechanisms by which serum G6PD activity increases in parallel with cancer progression. Forth, several confounders that could affect G6PD levels and gastric cancer progression were not fully considered. Baseline comorbidities such as diabetes mellitus, thyroid disorder, Helicobacter pylori infection, smoking status and chronic hepatitis are known to be associated with G6PD levels. However, these factors were not collected in our study. Nevertheless, most patients with underlying conditions were under regular management in their respective specialty clinics were in a stable, well-controlled state. Therefore, we expect that chronic conditions would not have caused substantial alterations in G6PD levels. Furthermore, because only 64 patients were included in this study, selection bias may have occurred, and the generalizability of the findings may be limited. As relatively few patients undergo G6PD level testing in clinical practice, the sample size was inevitably small. Further studies with larger cohorts are needed to validate our results and enhance the generalizability. In addition, there was a time interval between the measurement of G6PD levels and the assessment of cancer stage, raising the possibility that the cancer may have progressed by the time the G6PD level was measured, However, most patients at tertiary centers undergo outpatient follow-up and response evaluation every few months, resulting in a maximum interval of only several months. To minimize the limitation, we also confirmed whether any change in cancer stage occurred after the measurement of G6PD levels.

This study provides clinical evidence supporting the use of serum G6PD activity to distinguish between early-stage and advanced-stage gastric cancer. The G6PD activity test could aid not only in monitoring stage I–II gastric cancer patients during follow-up for potential recurrence as stage III–IV disease but also in assisting the initial clinical staging at diagnosis. Furthermore, given that the G6PD activity assay is portable, user-friendly, and yields rapid results, it is well suited for short-term follow-up assessments for gastric cancer patients.

## 5. Conclusions

This study demonstrates that serum G6PD activity may serve as a valuable biomarker for predicting the stage of gastric cancer. Patients with advanced-stage disease exhibited significantly higher G6PD activity compared with those in the early stage, and ROC analysis indicated fair diagnostic accuracy. These findings are consistent with previous reports showing increasing G6PD activity with cancer progression and decreased levels after treatment, suggesting that serum G6PD activity reflects tumor burden and could be clinically useful for disease staging and monitoring in gastric cancer.

## Figures and Tables

**Figure 1 cancers-17-03798-f001:**
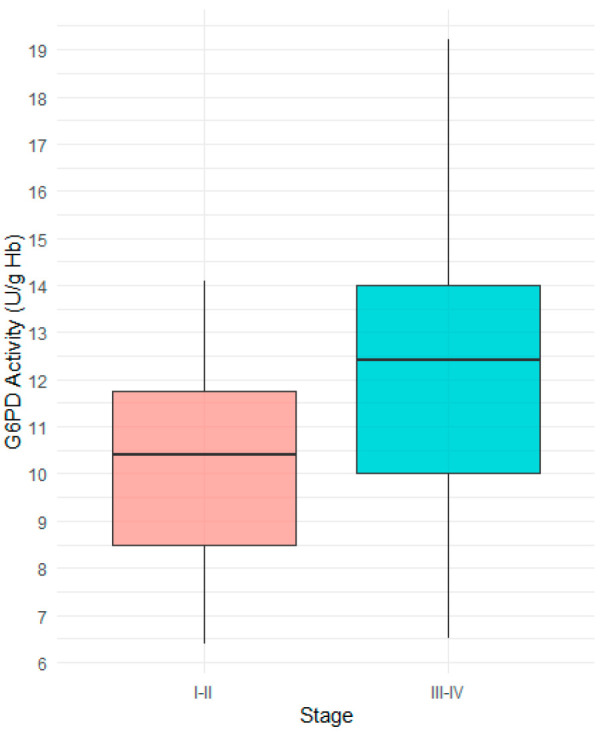
Comparison of G6PD activity between Stage I−II and Stage III−IV gastric cancer patients. The median G6PD activity was significantly higher in patients with Stage III−IV.

**Figure 2 cancers-17-03798-f002:**
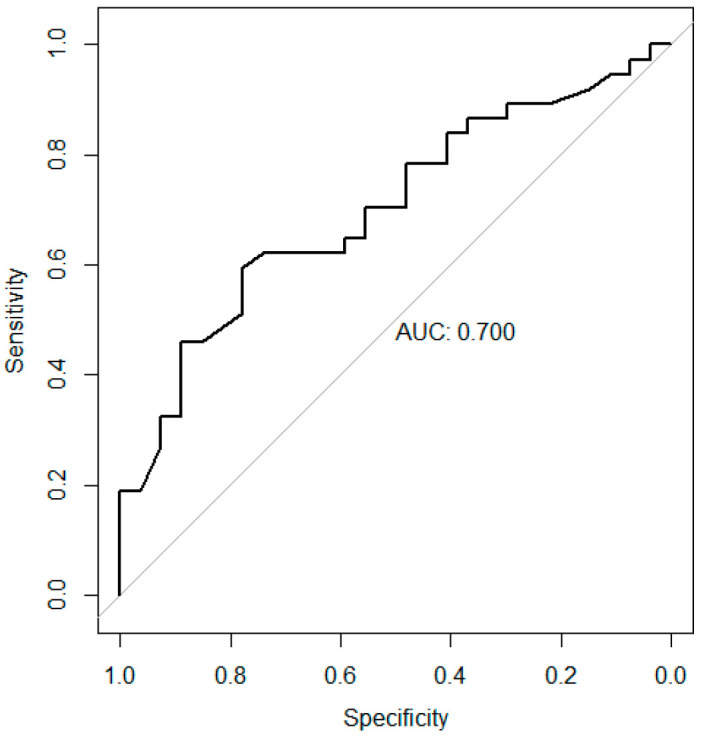
Receiver operating characteristic curve presenting the predictive ability of G6PD activity for Stage III–IV gastric cancer. The area under the curve was 0.70 (95% CI, 0.57–0.83).

**Table 1 cancers-17-03798-t001:** Comparison of baseline characteristics according to cancer stage (I–II vs. III–IV)

	Stage I–II (N = 27)	Stage III–IV (N = 37)	*p*-Value
Gender			0.035 *
Male	8 (29.6%)	22 (59.5%)	
Female	19 (71.4%)	15 (40.5%)	
Age (years)	57 (46–62)	59 (55–62)	0.127
G6PD activity (U/g Hb)	10.4 (8.5–11.8)	12.4 (10.0–14.0)	0.007 **

Data are shown as median (interquartile range, IQR) for continuous variables (Age, G6PD activity) and number of patients for categorical variable (gender). *p* value for gender was calculated using the chi-square test, while *p* values for age and G6PD activity were obtained using the Mann–Whitney U test. * *p* < 0.05, ** *p* < 0.01.

**Table 2 cancers-17-03798-t002:** Sensitivity, Specificity, PPV, NPV of G6PD activity for differentiating Stage III–IV gastric cancer patients at various cutoff values. The optimal cutoff value was 12.05 U/g Hb.

Cutoffs for G6PD Activity (U/g Hb)	Sensitivity	Specificity	PPV	NPV
10.05	0.73	0.48	0.66	0.57
11.05	0.65	0.59	0.69	0.55
12.05	0.59	0.78	0.79	0.58
13.05	0.41	0.89	0.83	0.52
14.05	0.19	0.96	0.88	0.46

## Data Availability

The datasets presented in this article are not publicly available due to ethical and legal restrictions, as they contain pseudonymized patient information. Requests to access the data may be considered on a case-by-case basis and should be directed to limmh12345846@gmail.com.

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
