# Peer review of "Serum Glucose-6-Phosphate Dehydrogenase Activity as a Biomarker for Gastric Cancer Stage Prediction"

_cancers, 2025, doi:10.3390/cancers17233798_

Round 1
Reviewer 1 Report
Comments and Suggestions for Authors
Overall, it’s a concise, clearly structured paper addressing an interesting and clinically relevant topic, but it has several methodological and interpretive weaknesses:
- The cross-sectional design and the adjustment for only age and sex weaken the strength of the association between G6PD activity and cancer stage. Important confounders such as diabetes, infection, or liver function were not considered.
- The study includes only 64 patients from a single center with no validation cohort. This limits generalizability and increases the risk of selection bias.
- The analysis of patients with and without bone metastasis appears unrelated to the main aim (stage prediction). If metastasis were to be evaluated all metastatic sites should have been analyzed or clearly defined as a secondary aim.
- With an AUC of 0.70 and moderate sensitivity/specificity. The discussion should be more balanced and less conclusive about clinical applicability.
- One author’s commercial involvement in G6PD device production and patent ownership. The manuscript should explain how independence of analysis was ensured.
Author Response
Reviewer #1:
We deeply thank you for taking the time to read our manuscript and to provide feedback. Your comments were highly insightful and helped us to improve the quality of our study. We addressed the concerns point by point in the following pages.
Point P1
The cross-sectional design and the adjustment for only age and sex weaken the strength of the association between G6PD activity and cancer stage. Important confounders such as diabetes, infection, or liver function were not considered.
(Response)
We are really grateful for reviewer’s comment. We also agree that several important confounders were not fully considered in our analysis. First, none of the participants in this study had any acute conditions, including acute infections, at the time serum G6PD levels were measured. We have added this information to the inclusion criteria. Second, baseline comorbidities such as diabetes or liver disease were not specifically collected in this study. We have therefore added this point to the limitations section. We sincerely appreciate reviewer’s insightful feedback regarding these important limitations of our study.
Line 102-105: “Patients who did not undergo serum G6PD activity test, had unknown gastric cancer stage, or were diagnosed with two or more types of cancer, or had any acute conditions including acute infections at the time of blood sampling were excluded.”
Line 333-339: “Forth, several confounders that could affect G6PD levels and gastric cancer progression were not fully considered. Baseline comorbidities such as diabetes mellitus, thyroid dis-order, Helicobacter pylori infection, smoking status and chronic hepatitis are known to be associated with G6PD levels. However, these factors were not collected in our study. Nevertheless, most patients with underlying conditions were under regular management in their respective specialty clinics were in a stable, well-controlled state. Therefore, we expect that chronic conditions would not have caused substantial alterations in G6PD levels.”
Point P2
The study includes only 64 patients from a single center with no validation cohort. This limits generalizability and increases the risk of selection bias.
(Response)
We sincerely appreciate the reviewer’s valuable comments. We also agree that the small number of participants limits the generalizability of our findings. Because only a limited number of patients undergo serum G6PD testing in routine clinical practice, the sample size of our study was inevitably small. We kindly ask for your understanding regarding this limitation. To improve external validity, we plan to conduct follow-up studies involving a larger population of cancer patients undergoing G6PD testing, which will help validate and generalize our results. We have added this point to the limitations section of the manuscript. We truly appreciate the reviewer’s insightful comments.
Line 340-344: “Furthermore, because only 64 patients were included in this study, selection bias may have occurred and the generalizability of the findings may be limited. As relatively few patients undergo G6PD level testing in clinical practice, the sample size was inevitably small. Further studies with larger cohorts are needed to validate our results and enhance the generalizability.”
Point P3
The analysis of patients with and without bone metastasis appears unrelated to the main aim (stage prediction). If metastasis were to be evaluated all metastatic sites should have been analyzed or clearly defined as a secondary aim.
(Response)
We sincerely apologize for the reviewer’s comment. The content related to metastasis, which is not relevant to the main focus of this study, was mistakenly included. We deeply regret this error. The irrelevant content has now been removed and the manuscript revised accordingly.
Editor Point P4
With an AUC of 0.70 and moderate sensitivity/specificity. The discussion should be more balanced and less conclusive about clinical applicability.
(Response)
We really appreciate for the reviewer’s advice regarding the clinical applicability of our study. We also agree that this aspect should be discussed in a balanced manner. While our findings suggest a potential role of serum G6PD activity as a biomarker, we acknowledge that there are certain limitations for its immediate application in clinical practice. We have added this point to the discussion section.
Line 301-308: “Considering that the model showed an AUC of 0.70 with moderate sensitivity and specificity, the result suggests a potential role of serum G6PD activity as a biomarker for gastric cancer stage, but the predictive performance is limited for immediate clinical ap-plication. Further validation in larger, independent cohorts will be necessary to confirm findings in this study and enhance generalizability. Therefore, while our study provides preliminary evidence supporting the association between serum G6PD levels and gastric cancer stage, additional studies are required to establish clinical utility of serum G6PD test.”
Editor Point P5
One author’s commercial involvement in G6PD device production and patent ownership. The manuscript should explain how independence of analysis was ensured.
(Response)
We are grateful for reviewer’s comment regarding conflict of interest. We also agree that it is important to describe this aspect in detail to ensure research ethics. We have added a statement clarifying that this study was conducted independently of any such conflict of interest and have describe the procedures followed to maintain independence.
Line 389-392: “Nevertheless, this study was conducted independently of any such conflicts of interest. First, G6PD level testing was performed for all cancer patients receiving vitamin C treatment. Second, the author (Chang-Hwan Yeom) with commercial involvement in the G6PD test did not participate in the final data analysis.”

Reviewer 2 Report
Comments and Suggestions for Authors
It is a privilege to review this valuable study. The work explores the potential of serum glucose-6-phosphate dehydrogenase (G6PD) activity as a biomarker for gastric cancer stage prediction, confirming its clinical value in distinguishing early and advanced stages through retrospective analysis and ROC curve analysis, providing a new perspective for non-invasive monitoring of gastric cancer. However, several methodological and conceptual aspects require refinement to enhance the rigor and generalizability of the research.
- Introduction:Clarify the potential interference of G6PD deficiency on serum G6PD activity, to rule out confounding factors in biomarker evaluation.
- Methods:Supplement detailed quality control parameters of the G6PD detection device, ensuring the reliability and reproducibility of test results.
- Introduction:The articles with PMID: 39323622, PMID: 39592900, PMID: 39950267, PMID: 39323626 and PMID: 37767800 are closely relevant to the background of this study, and citing them could help enrich the contextual discussion in the Introduction.
- Results:Analyze the correlation between G6PD activity and specific TNM sub-stages, to refine its predictive value for precise staging.
- Methods:Adjust for confounding factors such as Helicobacter pylori infection and smoking status, which are closely associated with gastric cancer progression.
- Discussion:Compare the diagnostic performance of G6PD activity with established gastric cancer biomarkers, to highlight its unique clinical advantages.
- Study Design:Perform gender-stratified analysis to address the gender imbalance between early and advanced stage groups, ensuring result robustness.
- Methods:Specify the time interval between G6PD testing and cancer staging, to avoid bias from disease progression during this period.
Author Response
Reviewer #1:
We deeply thank you for taking the time to read our manuscript and to provide feedback. Your comments were highly insightful and helped us to improve the quality of our study. We addressed the concerns point by point in the following pages.
Point P1
Introduction:Clarify the potential interference of G6PD deficiency on serum G6PD activity, to rule out confounding factors in biomarker evaluation.
(Response)
We are really grateful for reviewer’s comment regarding the potential confounding effect of G6PD deficiency on serum G6PD activity. To address this issue, patient with known G6PD deficiency were excluded from the study. In addition, participants with acute conditions, including acute infections, that could potentially affect G6PD levels were also excluded. We have incorporated all of these revisions into the manuscript. We sincerely appreciate the reviewer’s detailed advice regarding potential confounders.
Line 102-106: “Patients who did not undergo serum G6PD activity test, had unknown gastric cancer stage, or were diagnosed with two or more types of cancer, or had any acute conditions including acute infections at the time of blood sampling were excluded. In addition, patients with known G6PD deficiency or those diagnosed with G6PD deficiency were also excluded.”
Point P2
Methods:Supplement detailed quality control parameters of the G6PD detection device, ensuring the reliability and reproducibility of test results.
(Response)
We deeply appreciate the reviewer’s suggestion. The G6PD measurement device used in this study is an in vitro diagnostic medical device that has obtained both European CE certification and approval from the Korean Ministry of Food and Drug Safety (MFDS). Its quality control procedures and performance characteristics have been formally reviewed and validated by these regulatory authorities. The Instruction for Use (IFU) include regulator-approved performance data and validation results for key quality control parameters. These include repeatability and reproducibility (CV ranges of 3.81-18.05% and 3.92-16.61%, respectively), accuracy and correlation with a comparative product (correlation coefficient of 0.99), and analytical sensitivity parameters such as measurement range (0-300 U/dL), limit of detection (2 U/dL), limit of blank (0 U/dL), limit of quantitation (17 U/dL). We appreciate the opportunity to provide detailed information regarding the G6PD detection device. We have additionally incorporated this information into the Methods section of the manuscript.
Line 112-121: “Its quality control procedures and performance characteristics have been rigorously evaluated and validated by these regulatory authorities. According to the Instructions for Use (IFU), the device demonstrates regulator-approved performance and validated quality control parameters. Specifically, repeatability and reproducibility testing showed coefficients of variation ranging from 3.81% to 18.05% and 3.92% to 16.61%, respectively. Ac-curacy and correlation analyses revealed a strong agreement with a comparative product, with a correlation coefficient of 0.99. Analytical sensitivity was also verified, with a measurement range of 0-300 U/dL, a limit of detection of 2 U/dL, a limit of blank of 0 U/dL, and a limit of quantitation of 17 U/dL.”
Point P3
Introduction:The articles with PMID: 39323622, PMID: 39592900, PMID: 39950267, PMID: 39323626 and PMID: 37767800 are closely relevant to the background of this study, and citing them could help enrich the contextual discussion in the Introduction.
(Response)
We are really thankful for the reviewer’s insightful comment regarding the need to better integrate recent literature on the molecular and cellular mechanisms underlying gastric cancer. In response, we have revised the introduction to include a concise summary of key factors influencing gastric cancer progression, recurrence, and prognosis, such as tumor microenvironmental interactions, dysregulated signaling pathways, and relevant genetic alterations.
We also incorporated a statement emphasizing the complexity and heterogeneity of gastric cancer pathogenesis, which presents challenges in developing a single, reliable biomarker for disease staging and therapeutic monitoring. These additions provide stronger scientific context for the necessity of simple and non-invasive biomarkers, thereby reinforcing the rationale for investing serum G6PD activity in our study. We believe that the revisions enhance the overall clarity and depth of the introduction and we thank the reviewer again for the valuable suggestion.
Line 76-86: “The progression, recurrence, prognosis of gastric cancer are influenced by complex factors, including tumor microenvironment elements such as crosstalk between cancer-associated fibroblasts and myeloid cells, dysregulation of key signaling pathways like the PI3K/Akt/mTOR pathway, and specific genetic alterations such as the FAT4 gene mutation[11-14]. Additionally, factors such as histological subtype have also been reported to impact the prognosis and recurrence patterns of gastric cancer[15]. Because of complex pathogenic mechanisms and heterogeneous genetic mutations, developing a reliable single biomarker that can accurately predict the stage and monitor the therapeutic response of gastric cancer patients remains challenging. Therefore, it is essential to identify a simple, non-invasive, and reliable biomarker to provide efficient diagnosis strategies for gastric cancer patients.”
Editor Point P4
Results:Analyze the correlation between G6PD activity and specific TNM sub-stages, to refine its predictive value for precise staging.
(Response)
We really appreciate for the reviewer’s comment about the need for a more precise analysis. In this study, we collected information on cancer diagnosis and staging from official medical documents, including medical certificates, clinical records, and referral notes. However, depending on the hospital and the type of documents available, TNM staging could not be obtained in many cases. We fully agree that comparing G6PD levels according to specific TNM sub-stages is essential for more accurate stage prediction. In future studies, we will systematically collect TNM staging data to enable a more refined and precise analysis. We sincerely appreciate the reviewer for highlighting this important point.
Editor Point P5
Methods:Adjust for confounding factors such as Helicobacter pylori infection and smoking status, which are closely associated with gastric cancer progression.
(Response)
We are grateful for reviewer’s comment on the confounders that are not considered in this study. We fully agree that important variables related to gastric cancer, such as Helicobacter pylori infection status and smoking history, were not considered in our analysis. Although we adjusted for several variables including sex, age, and acute illness, we were unable to include smoking status, H. pylori infection, and other factors associated with G6PD levels such as diabetes mellitus and liver disease because these data were difficult to obtain from the available records. We have added this issue to the limitations section of the manuscript. We sincerely apologize for not being able to incorporate these factors in the present study, and we will make sure to include them in future research. We appreciate the reviewer’s insightful and important comment.
Line 333-339: “Forth, several confounders that could affect G6PD levels and gastric cancer progression were not fully considered. Baseline comorbidities such as diabetes mellitus, thyroid dis-order, Helicobacter pylori infection, smoking status and chronic hepatitis are known to be associated with G6PD levels. However, these factors were not collected in our study. Nevertheless, most patients with underlying conditions were under regular management in their respective specialty clinics were in a stable, well-controlled state. Therefore, we expect that chronic conditions would not have caused substantial alterations in G6PD levels.”
Editor Point P6
Discussion:Compare the diagnostic performance of G6PD activity with established gastric cancer biomarkers, to highlight its unique clinical advantages.
(Response)
We deeply appreciate for reviewer’s comment on the unique clinical advantages of G6PD activity compared to established gastric cancer biomarkers. In response, we have addressed this point in the discussion section. First, we described the diagnostic performance of other gastric cancer biomarkers, including miRNAs and circulating tumor cells (CTCs). Second, we compared their diagnostic performance with that of serum G6PD activity.
Biomarkers such as miRNAs and CTCs currently lack standardized measurement protocols. Moreover, while these biomarkers have been evaluated for their diagnostic utility in gastric cancer, their performance in predicting cancer stage has not been assessed. In contrast, G6PD activity offers the potential advantage of screening for cancer progression in gastric cancer patients. Finally, the G6PD activity test provides additional benefits in terms of accessibility and reproducibility, making it a practical tool for clinical application. We are thankful for reviewer for highlighting this important aspect.
Line 274-276: “In addition, miRNAs were evaluated solely for their diagnostic utility in gastric cancer, and the reported AUC values for the two miRNAs were 0.786, 0.769, respectively.”
Line 286-287: “Additionally, for the diagnosis of gastric cancer, circulating tumor cell (CTC) detection has been reported to show a sensitivity of 85.3% and a specificity of 90.3%[40].”
Line 293-297: “Moreover, for the diagnosis of gastric cancer, miRNAs and CTCs showed higher AUC, sensitivity, and specificity compared to serum G6PD levels. However, biomarkers that have been evaluated for their diagnostic performance in predicting gastric cancer stage are scarce, highlighting the advantage of serum G6PD levels as a potential tool to screen for disease progression in gastric cancer patients.”
Editor Point P7
Study Design:Perform gender-stratified analysis to address the gender imbalance between early and advanced stage groups, ensuring result robustness.
(Response)
We deeply appreciate for reviewer’s comment for suggesting a sensitivity analysis to ensure the robustness of our findings. We performed a gender-stratified analysis, comparing serum G6PD activity within each gender group and conducting ROC curve analyses. As a result, both men and women showed higher G6PD levels in stage III-IV compared with stage I-II. However, the difference was not statistically significant in women. In the ROC curve analysis, the AUC values were 0.830 for men, and 0.593 for women. We believe this is likely due to the small number of female patients (N=34), which limits the ability to detect statistically significant differences. In the future studies, we will aim to recruit a sufficient number of both male and female participants to allow robust gender-stratified analyses. The result of the gender-stratified analysis have been added to the Supplementary material.
Line 227-231: “In sensitivity analysis, the AUC values were 0.830 for men and 0.593 for women. This is likely attributable to the small number of female patients (N=34), which limits the ability to detect significant differences in G6PD activity or to adequately assess its diagnostic performance. Future studies with a sufficiently large number of both male and female participants will be necessary to validate the findings.”
Supplementary file
Editor Point P8
Methods:Specify the time interval between G6PD testing and cancer staging, to avoid bias from disease progression during this period.
(Response)
We deeply appreciate reviewer’s comment regarding the accuracy of cancer staging. The patient included in this study were those receiving diagnosis and treatment at a tertiary hospital, undergoing regular outpatient follow-up or response evaluation. Therefore, the time interval between G6PD testing and the confirmation of the final cancer staging was within three months. We have also described as a study limitation the possibility that disease progression from stage I-II to stage III-IV may have occurred during the period between the confirmation of the final cancer stage and the subsequent G6PD testing. Furthermore, to minimize this limitation, we also reviewed whether any change in cancer stage occurred after the measurement of G6PD levels.
Line 344-350: “In addition, there was a time interval between the measurement of G6PD levels and the assessment of cancer stage, raising the possibility that the cancer may have progressed by the time the G6PD level was measured, However, most patients at tertiary center undergo outpatient follow-up and response evaluation every few months, resulting in a maximum interval of only several months. To minimize the limitation, we also confirmed whether any change in cancer stage occurred after the measurement of G6PD levels.”

Round 2
Reviewer 1 Report
Comments and Suggestions for Authors
All my previous comments have been fully addressed.